# Rifaximin Ameliorates Loperamide-Induced Constipation in Rats through the Regulation of Gut Microbiota and Serum Metabolites

**DOI:** 10.3390/nu15214502

**Published:** 2023-10-24

**Authors:** Mei Luo, Peiwei Xie, Xuehong Deng, Jiahui Fan, Lishou Xiong

**Affiliations:** Department of Gastroenterology and Hepatology, The First Affiliated Hospital of Sun Yat-sen University, Guangzhou 510080, China; 18070591383@163.com (M.L.); xiepw5@mail2.sysu.edu.cn (P.X.); dxh13068698775@163.com (X.D.); 19878808964@163.com (J.F.)

**Keywords:** rifaximin, loperamide, constipation, 5-HT, gut microbiota, metabolites, bile acids

## Abstract

Structural changes in the gut microbiota are closely related to the development of functional constipation, and regulating the gut microbiota can improve constipation. Rifaximin is a poorly absorbed antibiotic beneficial for regulating gut microbiota, but few studies have reported its effects on constipation. The purpose of this study was to investigate the effect of rifaximin on loperamide-induced constipation in SD rats. The results showed that rifaximin improved constipation by increasing serum 5-HT, SP, and the mRNA expression of AQP3, AQP8, and reducing the mRNA expression of TLR2 and TLR4. In addition, rifaximin could regulate the gut microbiota of constipated rats, such as increasing the potentially beneficial bacteria *Akkermansia muciniphila* and *Lactobacillus murinus*, reducing the *Bifidobacterium pseudolongum*. According to metabolomics analysis, many serum metabolites, including bile acids and steroids, were changed in constipated rats and were recovered via rifaximin intervention. In conclusion, rifaximin might improve loperamide-induced constipation in rats by increasing serum excitatory neurotransmitters and neuropeptides, modulating water metabolism, and facilitating intestinal inflammation. Muti-Omics analysis results showed that rifaximin has beneficial regulatory effects on the gut microbiota and serum metabolites in constipated rats, which might play critical roles in alleviating constipation. This study suggests that rifaximin might be a potential strategy for treating constipation.

## 1. Introduction

Functional constipation (FC) is a common problem with an increasing incidence rate and it negatively affects quality of life and generates major health care costs for patients [1,2]. The general symptoms of FC include reduced bowel movement frequency (less than three times/week), hard stools, difficulty in defecating, a sense of incomplete evacuation after defecation with or without abdominal pain, and bloating [3]. The pathogenesis of FC is complex, and its pathogenesis is still unclear. Our previous study has shown that many patients with FC were dissatisfied with their treatment, with ineffective treatment being the most common reason [4]. Therefore, new strategies are needed to address constipation.

Accumulating studies have shown a specific correlation between the structural changes in gut microbiota and the development of FC [5,6,7,8]. Khalif et al. [5] demonstrated that the levels of Bifidobacteria, Lactobacilli, Bacteroides, and Clostridium species were decreased in FC patients, while the levels of Enterobacteriaceae, Staphylococcus aureus, and fungi were increased. Zoppi et al. [9] showed that, compared to the healthy controls (HC), FC patients had a significantly higher level of Clostridium and Bifidobacterium species in the fecal samples. The main characteristics of the gut microbiota in FC are a relative decrease in beneficial bacteria and species richness and an increase in potential pathogens [10]. Gut microbiota can regulate gut functions through the metabolites of bacterial fermentation, among which 5-hydroxytryptamine (5-HT), short-chain fatty acids (SCFAs), methane, and bile acids (BAs) occupied more important positions [11,12,13,14]. And then these metabolites can influence gut sensation, secretion, and motility. Therefore, the regulation of the gut microbiota may be a new strategy for the treatment of FC.

Rifaximin is a broad-spectrum antibiotic with activity against both Gram-positive and Gram-negative, anaerobic and aerobic bacteria [15]. It is a poorly absorbed oral antibiotic with a high safety profile due to its low systemic absorption [16]. Rifaximin is able to inhibit the bacterial RNA synthesis through binding the β-subunit of bacterial DNA-dependent RNA polymerase, resulting in the inhibition of bacterial protein synthesis [17]. Rifaximin could inhibit bacterial overgrowth in the small intestine by modifying the intestinal microecology [18]. Therefore, rifaximin has been used to treat small intestinal bacterial overgrowth (SIBO), a heterogeneous syndrome characterized by an increased number and/or abnormal type of bacteria in the small bowel [19,20]. Jian et al. [21] showed that rifaximin ameliorated non-alcoholic steatohepatitis in mice through regulating gut microbiome-related BAs. Hong et al. [22] demonstrated that rifaximin treatment reduced the abundance of Prevotellaceae UCG-001 and increased the abundance of Bacteroides, Muribaculum, and Lachnospiraceae UCG-001 in Parkinson’s disease mice. In addition, rifaximin treatment attenuated serum interleukin-1β, interleukin-6 and tumor necrosis factor-α, claudin-5, and occludin. Studies have also shown that rifaximin could actively regulate the composition of the gut microbiome by promoting the growth of beneficial bacteria, such as Bifidobacteria and Lactobacilli [23,24]. Similarly, disorders in the gut microbiota are considered a pathogenesis of constipation, and a few studies showed that rifaximin could improve constipation [25]. However, there are currently few studies on the use of rifaximin to treat constipation, and the efficacy and mechanism of rifaximin in improving constipation are unclear.

In this study, we used rifaximin to intervene in male SD rats with loperamide-induced constipation and explored whether rifaximin could improve constipation-related physiochemical indices. Meanwhile, we investigated the effects of rifaximin on gut microbiota and serum metabolites in constipated rats and analyzed the role of rifaximin in the pathogenesis of constipation and the potential mechanism.

## 2. Materials and Methods

### 2.1. Animal Experiment

Twenty-one six-week-old (weight 180–200 g) male Sprague Dawley (SD) rats were purchased from Zhuhai BesTest Bio-Tech Co, Ltd. (Zhuhai, China). All rats were allowed to adapt to the laboratory conditions for one week before the experiment. The rats were housed in cages under standard conditions at a room temperature of 25 °C ± 2 °C and humidity of 50% ± 5% with a 12 h (h) light–dark cycle (07:30–19:30). They were fed standard food and had free access to food and water during the period of the experiment. The experiments were approved by the Animal Care Review Committee of Sun Yat-sen University.

The constipation model of rats was induced by loperamide hydrochloride (L4762, Sigma–Aldrich, St. Louis, MO, USA) according to the previous study [26,27]. These rats were randomly divided into three groups of seven rats each: control group (CTR group), loperamide group (LOP group), and rifaximin group (RIF group). The rats in the LOP and RIF groups were injected subcutaneously with 5 mg kg^−1^ of body weight loperamide in 0.9% saline solution twice daily at 09:00 am and 5:00 pm for 14 days, while the rats in the CTR group were injected with 0.9% saline. Rifaximin (R830175, Macklin, Shanghai, China) was dissolved in 0.9% saline to a final concentration of 50 mg/mL. The rats in the RIF group were treated with rifaximin (75 mg kg^−1^ body weight) by gavage administration one hour after LOP was injected when the rats in the CTR and LOP groups were administered 0.9% saline.

### 2.2. Physiological Indices for the Rats 

#### 2.2.1. Body Weight Changes and Food Intake

The body weight and food intake of all rats were weighed with precision electronic balance (METTLER TOLEDO, ME2002E/02) at 8:00 a.m. every day from the first day of the experiment to the last day. The food intake was measured according to the following equation: Food intake (g) = the weight of food on the day before − the weight of food on the day

#### 2.2.2. Fecal Parameters

The number of feces excreted by the rats was measured between 9:00 a.m. each day and 9:00 a.m. the following day during the experiment. To calculate the fecal water content, fresh feces of each rat were collected and immediately weighed and then dried at 60 °C for 24 h to obtain the fecal dry weights. The fecal water content was calculated according to the following equation:$$Fecal\;water\;content\;=\;\frac{fecal\;wet\;weight\;—fecal\;dry\;weight}{fecal\;wet\;weight}\;\times \;100\%$$

#### 2.2.3. Feces, Blood, and Colon Tissue Collection

After the last administration, fresh feces from all rats were collected in sterile tubes and immediately frozen and stored at −80 °C. Then, all rats were fasted for 12 h but allowed free access to water. Each rat was administered 1.0 mL activated charcoal solution (an aqueous suspension of 5% charcoal and 10% gum arable) by gavage. Thirty minutes later, the rats were anesthetized by injecting sodium pentobarbital. Blood samples were collected and placed at room temperature for two hours, centrifuged at 3000 rpm for 10 min to obtain serum, and stored at −80 °C until analysis. Then, the rats were euthanized and dissected. The small intestines were removed from the stomach to the ileum, and the distance covered by the charcoal and the total length of the small intestine were measured. After that, the intestinal segment was removed from the cecum to the anus and washed with PBS, dividing the tissue into two parts: one was immediately frozen and stored at −80 °C until analysis, and the other was kept in a formalin solution for paraffin embedding.

#### 2.2.4. Intestinal Propulsive Rate

For each rat, the intestinal propulsive rate was calculated as the percentage of the distance moved by the activated charcoal relative to the total length of the small intestine. The intestinal propulsive rate was calculated according to the following equation:$$Intestinal\;propulsive\;rate\;=\;\frac{distance\;moved\;by\;the\;charcoal}{total\;length\;of\;the\;small\;intestine}\;\times \;100\%$$

### 2.3. Hematoxylin and Eosin Stain (H&E)

After being fixed in 4% paraformaldehyde, the colon tissues of rats were embedded in paraffin and sliced, followed by histopathological examination by hematoxylin and eosin (H&E) staining. Then, the pathological changes in colon tissues were examined using a microscope.

### 2.4. Enzyme-Linked Immunosorbent Assay (ELISA)

The serum was removed from the −80 ℃ refrigerator and placed at room temperature until melted. The concentrations of serum 5-HT, SP, VIP, and MTL were measured with corresponding enzyme-linked immunosorbent assay (ELISA) kits (Mei Mian) following the manufacturer’s instructions.

### 2.5. RNA Extraction and Real-Time Quantitative Polymerase Chain Reaction (RT–qPCR)

A real-time quantitative polymerase chain reaction (RT–qPCR) detection system was used to quantify the expression of mRNA. Total RNA in the colon tissues of rats was extracted using TRIzol reagent (Invitrogen, Waltham, MA USA). MonScript™ RTIII All-in-One Mix with dsDNase was used for RNA reverse transcription, the RNA reverse transcription reaction mixture included 1 μg template RNA, 4 μL 5 × RTIII All-in-One Mix, 1 μL dsDNase, and Nuclease-Free Water to a final volume of 20 µL. MonAmp™ SYBR^®^ Green qPCR Mix (None ROX) was used for the quantitative PCR analysis of gene expression, the PCR reaction mixture including 10 μL SYBR^®^ Green qPCR Mix (None ROX), 0.4 μL forward primer, 0.4 μL reverse primer, 200 ng template DNA, and Nuclease-Free Water to a final volume of 20 µL. Then, the mRNA gene expression levels of inflammatory cytokines (TLR2 and TLR4) and aquaporins (AQP3 and AQP8) were quantified. The relative level of the target mRNA was normalized to the β-actin level, and the primer sequences were as follows: TLR2-sense, 5′-TGGAGGTCTCCAGGTCAAATC-3′; TLR2-antisense, 5′-ACCAGCAGCATCACATGACA-3′; TLR4-sense, 5′-GATCTGAGCTTCAACCCCCT-3′; TLR4-antisense, 5′-TTGTCTCAATTTCACACCTGGA-3′; AQP3-sense, 5′-GCTGCTGTGCCTATGAACTGA-3′; AQP3-antisense, 5′-CTTCTTGGGTGCTGGGATTGT-3′; AQP8-sense, 5′-GGGATCTCTGGAGCCTGCATG-3′; AQP8-antisense, 5′-CTGCTGCTGTCAGAGTGGCTC-3′; β-actin-sense, 5′-GATTACTGCCCTGGCTCCTAG-3′; and β-actin-antisense, 5′-GAAAGGGTGTAAAACGCAGCTC-3′. The results of target mRNA levels were calculated using the 2^−∆∆Ct^ method.

### 2.6. 16S rRNA High-Throughput Sequencing of Fecal Microbiota

Fecal microbial DNA from rat feces was extracted using an E.Z.N.A.^®^ soil Kit (Omega Biotek, Norcross, GA, USA) according to the manufacturer’s protocols. The V3-V4 hypervariable regions were amplified with primers 338F-806R using a thermocycler PCR system (GeneAmp 9700, ABI, Natick, MA, USA). The PCR reaction mixture including 4 μL 5× Fast Pfu buffer, 2 μL 2.5 mM dNTPs, 0.8 μL each primer (5 μM), 0.4 μL Fast Pfu polymerase, 10 ng of template DNA, and ddH_2_O to a final volume of 20 µL. PCR amplification was conducted using the following procedure: 3 min of denaturation at 95 °C, 27 cycles of 30 s at 95 °C, 30 s for annealing at 55 °C, and 45 s for elongation at 72 °C, and a final extension at 72 °C for 10 min. Purified amplicons were pooled in equimolar amounts and paired-end sequenced on an Illumina MiSeq platform (Major Bio-Pharm Technology, Shanghai, China) according to the standard protocols by Majorbio Bio-Pharm Technology Co., Ltd. (Shanghai, China).

After sequencing, raw fastq files were demultiplexed, quality-filtered using Trimmomatic and merged using FLASH. Operational taxonomic units (OTUs) were clustered with a 97% similarity cutoff using UPARSE, and chimeric sequences were identified and removed using UCHIME. The taxonomy of each 16S rRNA gene sequence was analyzed using the RDP Classifier algorithm against the Silva (SSU123) 16S rRNA database using a confidence threshold of 70%.

Alpha-diversity analyses of community diversity parameters (Chao index, Shannon index) were calculated using Mothur 1.30.2 software. Principal coordinate analysis (PCoA) is a nonconstrained method of data dimensionality reduction analysis that can be used to analyze the similarity or difference in the composition of bacterial communities. A Venn diagram was used to count the number of shared and unique bacteria in multiple groups or samples. Linear discriminant analysis effect size (LEfSe) was used to quantify the differential bacteria between groups. PICRUSt2 analysis (KEGG level) was used to predict the functional profiling of microbial communities.

### 2.7. Metabolomics Analysis

Liquid chromatography–tandem mass spectrometry (LC–MS/MS) analysis was used to analyze the serum metabolites. The specific analytical methods are provided in the Appendix A.

### 2.8. Statistical Analysis

Statistical differences in data were assessed using Student’s *t* test (statistical difference was compared among two groups and data followed normal distribution), the Kruskal–Wallis test (statistical difference was compared among two groups and data did not follow normal distribution), and one-way ANOVA (statistical difference was compared among three groups). SPSS 23.0 and GraphPad Prism 8.0 were used for data analysis and drawing. All data are expressed as the mean ± standard deviation (SD) (the results were presented in graphic form) or mean (the results were presented in tabular form), and a *p* value < 0.05 was considered to be statistically significant.

## 3. Results

### 3.1. Rifaximin Ameliorates Physiological Indices in Constipated Rats

Body weight, food intake, fecal number, fecal water content, and gastrointestinal transit rate are the five physiological indices for evaluating constipation in rats. The results showed that rifaximin intervention suppressed the decline in body weight of constipated rats compared to control rats (Figure 1A). During the experiment, the daily food intake of the rats in the LOP group was less than that of the CTR and RIF groups (Figure 1B). As shown in Figure 1C,D, the fecal number and fecal water content in the LOP group were significantly lower than those in the CTR and RIF groups. After measuring the distance traveled by activated charcoal and calculating the intestinal propulsive rate, the intestinal propulsive rate in the LOP group was significantly reduced compared to that in the CTR group, indicating that the intestinal motility in constipated rats was slower than that in control rats and that rifaximin accelerated the intestinal motility of constipated rats (Figure 1E). The H&E staining method was used to observe the pathological changes in colon tissues in rats. As shown in Figure 1F, there were obvious structural destructions of colon tissues in the LOP group, including mucosal epithelial cell necrosis and abscission, the muscularis mucosa becoming thinner, and goblet cells decreasing. Rifaximin intervention reduced the structural destruction of colon tissues in rats.

### 3.2. Rifaximin Affects Serum Neurotransmitters, Neuropeptides, and mRNA Expression of Inflammatory Cytokines and Aquaporins in Colon Tissues

To evaluate the effects of rifaximin on neurotransmitters and neuropeptides in rats, we used ELISA kits to measure the concentrations of 5-HT, SP, MTL, and VIP in the serum of rats. As shown in Figure 2A,B, although there were no significant differences in the concentrations of 5-HT and SP between the CTR and LOP groups, the concentrations of 5-HT and SP in the RIF group were significantly increased compared to those in the LOP group. In addition, there were no significant differences in the concentrations of VIP and MTL between the three groups (Figure 2C,D). Next, we explored the effect of rifaximin on water metabolism and intestinal inflammation in the colon of constipated rats using RT–qPCR. As shown in Figure 2E,F, the mRNA levels of AQP3 and AQP8 were low in constipated rats compared to control rats, and rifaximin intervention increased the expression of AQP3 and AQP8. In addition, inflammatory cytokines, including TLR2 and TLR4, were overexpressed in the constipated rats, and rifaximin attenuated this overexpression (Figure 2G,H). Then, we analyzed the correlation between AQP, AQP8,5-HT, SP, fecal water content, and gastrointestinal transit rate using Spearman correlation analysis in a heatmap. AQP3 and AQP8 showed significantly positive correlations with fecal water content (Figure 2I), which indicates that the low expression of AQP3 and AQP8 in constipated rats might affect the water metabolism of the colon lumen, increase water reabsorption from the luminal side to the vascular side in the intestine, and reduce the water content in the feces. Additionally, 5-HT showed a significantly positive correlation with the gastrointestinal transit rate, which is consistent with most studies’ findings that the concentration of 5-HT is closely related to colon motility.

### 3.3. Rifaximin Regulates the Structures of Gut Microbiota in Constipated Rats

After the experiments, fresh feces of all rats were collected to sequence the V3–V4 region using the 16S rRNA gene to analyze the effect of rifaximin intervention on the gut microbiota of constipated rats. After demultiplexing and quality-filtering the original sequencing data, 2211 OTUs were obtained. As shown in Figure 3A,B, in the α-diversity analysis, the Chao index in the RIF group was lower than that in the CTR and LOP groups, indicating that rifaximin might inhibit the growth of some potentially pathogenic bacteria, and there were no significant differences in the Shannon index between the CTR, LOP, and RIF groups. In the β-diversity analysis, the CTR and LOP groups were separated in the PCoA plot, while RIF was separated between them (Figure 3C), which showed that there were significant differences in the gut microbiota between the CTR and LOP groups and that rifaximin intervention could decrease the disparities. We further analyzed the differences in the abundance of bacterial communities between the three groups at the levels of phylum, genus, and species (Figure 3D–H). A total of 15 phyla, 262 genera, and 451 species were found in the feces of the three groups, and the shared and unique fecal bacterial communities between the three groups at the phylum, genus, and species levels are shown in Figure 3D–F. The results of LEfSe analysis showed bacterial communities with significant differences in abundance (cladograms, LDA > 3.5, Figure 3G) and dominant bacterial communities (histogram of LDA value distribution, LDA > 3.5, Figure 3H) in the CTR, LOP, and RIF groups.

Then, we analyzed the differences in gut microbiota among the three groups at the phylum, genus, and species levels. We mainly focused on where the gut microbiota was significantly affected after rifaximin intervention. At the phylum level, the abundance of Verrucomicrobiota in the RIF group was significantly increased, while Actinobacteriota, Patescibacteria, Proteobacteria, and unclassified_k__norank_d__Bacteria were reduced compared to the LOP group (Figure 4A). At the genus level, after rifaximin intervention, the abundances of 50 microbial communities were altered compared to those in LOP (Appendix A). Among the top 10 microbial communities in abundance, *Akkermansia* and *Ruminococcus_gauvreauii_group* were significantly increased, while *Romboutsia*, *Dubosiella*, *Clostridium_sensu_stricto_1*, *Allobaculum*, *Bifidobacterium*, *norank_f__norank_o__Clostridia_UCG-014*, *Coriobacteriaceae_UCG-002*, and *Faecalibaculum* were significantly reduced compared to LOP (Figure 4B). At the species level, the abundances of 82 microbial communities in the RIF group were altered compared to those in the LOP group (Appendix A). Among the top 10 microbial communities in abundance, *Akkermansia muciniphila*, *Lactobacillus murinus* and *uncultured_organism_g__Ruminococcus_gauvreauii_group* were significantly increased, while *Romboutsia ilealis*, *uncultured_bacterium_g__Dubosiella*, *uncultured_bacterium_g__Clostridium_sensu_stricto_1*, *Bifidobacterium pseudolongum*, *unclassified_g__norank_f__norank_o__Clostridia_UCG-014*, *uncultured_bacterium_g__Coriobacteriaceae_UCG-002*, and *Faecalibaculum rodentium* were significantly reduced compared to LOP (Figure 4C). These results indicated that rifaximin might improve constipation by regulating gut microbiota, thereby accelerating gastrointestinal motility. 

Spearman correlation analysis was used to analyze the correlations between gut microbiota and neurotransmitters, neuropeptides, aquaporins, and inflammatory cytokines. As shown in Figure 4D, 13 bacteria (*Akkermansia muciniphila*, *Uncultured_bacterium_g_Eubacterium_nodatum_group*, and *uncultured_bacterium_g_Enterorhabdus*, etc.) showed positive correlations with AQP3 and AQP8, while *Bifidobacterium_pseudolongum*, *uncultured_bacterium_g__Clostridium_sensu_stricto_1*, and *uncultured_bacterium_g__Coriobacteriaceae_UCG-002* showed negative correlations with AQP3 and AQP8. In addition, *Akkermansia muciniphila*, *uncultured_rumen_bacterium_g__norank_f__norank_o__RF39*, and *uncultured_bacterium_g__Eubacterium_ruminantium_group* showed positive correlations with 5-HT and SP, which were increased after rifaximin intervention. *Bifidobacterium pseudolongum*, *uncultured_bacterium_g_Clostridium_sensu_stricto_1*, *uncultured_bacterium_g_Coriobacteriaceae_UCG-002*, *unclassified_g_Allobaculum*, and *uncultured_bacterium_g_norank_f_norank_o_Clostridia_UCG-014* showed positive correlations with TLR2 and TLR4.

### 3.4. Functional Prediction Analysis of Gut Microbiota

Based on 16S rRNA high-throughput sequencing data and combined with Kyoto Encyclopedia of Genes and Genomes (KEGG) databases, we used PICRUSt2 analysis to perform a functional prediction analysis on the gut microbiota discovered in this study (Figure 5). As shown in Figure 5A, according to the predicted results of PICRUSt2, the gut microbiota in our study primarily participated in metabolic activity. Combining deeper (Level 2) KEGG data information, it can be confirmed that the functions of gut microbiota are mainly related to the metabolic pathways of carbohydrate metabolism, amino acid metabolism, energy metabolism, metabolism of cofactors and vitamins, nucleotide metabolism, and lipid metabolism (Figure 5B). Further prediction (Level 3) results showed a close relationship between gut microbiota and the metabolic pathways of microbial metabolism in diverse environments, in the biosynthesis of amino acids, in carbon metabolism, purine metabolism, and amino sugar and nucleotide sugar metabolism (Figure 5C). These results indicate a close relationship between gut microbiota and host metabolic activity.

### 3.5. Rifaximin Affects Serum Metabolites in Constipated Rats

To analyze the effects of rifaximin on metabolites in the serum of constipated rats, we collected serum samples from all rats for untargeted metabolomic analysis through LC–MS/MS. A total of 1982 metabolites were detected in serum samples in the three groups. Based on the results of the principal component analysis (PCA) and partial least squares-discriminant analysis (PLS-DA), the serum metabolites showed significant differences between the three groups (Figure 6A,B). As shown in the results of the volcano map, compared with the CTR group, 381 serum metabolites were upregulated and 178 metabolites were downregulated in the LOP group (Figure 6C). In the RIF group, 156 metabolites were upregulated and 370 were downregulated compared to the LOP group (Figure 6D). To further determine the metabolic pathway of rifaximin in improving constipation in rats, we conducted metabolic pathway analysis on differentially abundant metabolites between groups and found that bile acid and steroid metabolic pathways were significantly different among the three groups. As shown in Figure 6E,F, metabolic pathways, including primary bile acid biosynthesis, bile secretion, steroid hormone biosynthesis, and steroid biosynthesis, were involved in the development of constipation and intervention with rifaximin in constipation.

We further analyzed the differentially abundant metabolites in the primary bile acid biosynthesis, bile secretion, steroid hormone biosynthesis, and steroid biosynthesis metabolic pathways between the three groups (Figure 7A–D and Appendix A). In the primary bile acid biosynthesis metabolic pathway, the relative levels of 7alpha,26-dihydroxycholest-4-en-3-one, glycochenodeoxycholic acid (GCDCA), chenodeoxycholic acid, 7alpha-hydroxy-3-oxo-4-cholestenoic acid, 3,7-dihydroxycoprostanic acid, and taurochenodeoxycholic acid were upregulated in the LOP group compared to the CTR group, and most of these changes could be suppressed after rifaximin intervention (Figure 7A). In the bile secretion metabolic pathway in the LOP group, glycochenodeoxycholic acid, chenodeoxycholic acid and taurochenodeoxycholic acid were upregulated, which was similar to the primary bile acid biosynthesis metabolic pathway. In addition, deoxycholic acid, leukotriene C4, and lithocholic acid were upregulated in the LOP group. At the same time, thyroxine, L-carnitine, carnitine, liothyronine and tetracycline were downregulated when compared to the CTR group, and most of these changes could be suppressed after rifaximin intervention (Figure 7C). In addition to the two metabolic pathways related to bile acids mentioned above, some serum metabolites in steroid biosynthesis and steroid hormone biosynthesis metabolic pathways were also found to be significantly altered. In steroid hormone biosynthesis metabolic pathways, androstanedione, tetrahydrocortisol, androstanolone, testosterone glucuronide and 5beta-dihydrocorticosterone were upregulated in the LOP group, while estrone and estrone glucuronide were downregulated (Figure 7B). In steroid biosynthesis metabolic pathways, Vitamin D3, Calcitriol and Zymosterol were upregulated in the LOP group (Figure 7D). Interestingly, we found that the concentrations of loperamide, which was used to induce constipation, were downregulated in the RIF group compared to the LOP group (Figure 7E). Most of these changes in the LOP group could be suppressed by rifaximin intervention, indicating that rifaximin could alleviate constipation by regulating the relative levels of serum metabolites. 

### 3.6. Correlations between Gut Microbiota and the Metabolites Involved in the Primary Bile Acid Biosynthesis, Bile Secretion, Steroid Hormone Biosynthesis, and Steroid Biosynthesis Metabolic Pathways

As mentioned above, some serum metabolites changed in metabolite pathways were found to be involved in the development of constipation, and the intervention of rifaximin could suppress these changes. It is well known that the gut microbiota can affect serum metabolites. Therefore, we used Spearman correlation analysis to analyze the correlations between gut microbiota and changed serum metabolites above. As shown in Figure 8, 17 bacterial (*Faecalibaculum rodentium*, *Bifidobacterium pseudolongum*, and *unclassified_g__Allobaculum*, etc.) species levels showed positive correlations with 18 serum metabolites (glycochenodeoxycholic acid, chenodeoxycholic acid, and taurochenodeoxycholic acid, etc.). In comparison, seven bacteria (*gut_metagenome_g__Lactobacillus*, *uncultured_bacterium_g__Lactobacillus*, and *Lactobacillus murinus*, etc.) were negatively correlated with these metabolites. In addition, 16 bacteria (*Faecalibaculum rodentium*, *Bifidobacterium pseudolongum*, and *unclassified_g__Allobaculum*, etc.) were negatively correlated with thyroxine, L-carnitine, carnitine, liothyronine, tetracycline, estrone, and estrone glucuronide. Spearman correlation analysis confirmed the interrelations between the metabolites and gut microbiota in this study.

### 3.7. Rifaximin Affects Serum Bas in Constipated Rats

We can see from the above findings that Bas play an important role in the development of constipation. There were 28 BAs detected by untargeted metabolomics in the three groups. We found that, compared with the CTR group, the majority of BAs showed a decreasing trend in the CTR group (Table 1). After rifaximin intervention, most increased BAs in the LOP group were inhibited (Table 2), which indicates that rifaximin intervention might reduce the reabsorption of BAs.

### 3.8. Correlations between Gut Microbiota and BAs

As shown in Figure 9, 12 bacterial (*Faecalibaculum rodentium*, *Bifidobacterium pseudolongum*, and *unclassified_g__Allobaculum*, etc.) species levels showed positive correlations with 23 BAs (deoxycholic acid, murideoxycholic acid, and apocholic acid, etc.) and negative correlations with the others. In comparison, five bacteria (*gut_metagenome_g__Lactobacillus*, *Lactobacillus murinus*, and *Ruminococcus_sp._N15. MGS-57_g__Ruminococcus*, etc.) were negatively correlated with the 23 BAs mentioned above but positively correlated with others. Interestingly, only the five bacteria negatively correlated with the 23 BAs were upregulated in the RIF group compared to the LOP group.

## 4. Discussion

Previous studies have shown that changes in gut microbiota structures are associated with the development of constipation [28,29] and that regulating gut microbiota can improve constipation [30,31,32]. Rifaximin, a poorly absorbed antibiotic, could regulate the composition of the gut microbiome [23,24]. Ghoshal et al. [25] demonstrated that rifaximin improves constipation by reducing breath methane and colon transit time. However, they did not investigate the effects of rifaximin on the gut microbiota and metabolites in patients with constipation. In our study, we investigated the effects of rifaximin on constipation-related symptoms in constipated rats. Also, we explored the effects of rifaximin on the gut microbiota and metabolites of constipation rats through multi-omics analysis. In this study, we induced constipation using loperamide in an SD rat model and intervened with rifaximin to explore whether rifaximin could alleviate constipation by regulating gut microbiota. The results suggested that rifaximin could improve loperamide-induced constipation in rats, including increased body weight, bowel movement frequency, daily food intake, fecal water content, and intestinal propulsive rate. We further found that rifaximin intervention could increase serum excitatory neurotransmitters, neuropeptides, and the mRNA expression of colon mucosal aquaporins and reduce the mRNA expression of inflammatory cytokines in colon tissues. In addition, after intervention with rifaximin, there were specific changes in the gut microbiota and serum metabolites in constipated rats.

Damaged colon mucosa could weaken intestinal peristalsis [33]. The results of H&E staining in our study showed that the colon tissues in the LOP group displayed mucosal epithelial cell necrosis and abscission, muscularis mucosa were thinner and goblet cells were decreased. However, rifaximin intervention reduced the structural destruction of colon tissues. The mRNA expression of the proinflammatory cytokines TLR2 and TLR4 decreased significantly after rifaximin intervention, which could reduce the destruction of tight junction proteins and intestinal permeability [34]. These results indicated that rifaximin could maintain the colonic mucosal barrier and relieve intestinal inflammation.

Aquaporins are expressed in the mucosal epithelial cells of the colon in animals and are closely related to the pathogenesis of constipation [35,36]. In this study, the mRNA expression levels of AQP3 and AQP8 were decreased in the colonic mucosa in loperamide-induced constipation rats, and rifaximin significantly increased the expression of AQP3 and AQP8. The results of the Spearman correlation coefficient proved that AQP3 and AQP8 showed strong positive correlations with fecal water content. Similarly, Shi and Kim et al. [37,38] demonstrated that upregulating the expression of AQP3 and AQP8 could improve loperamide-induced constipation. These results indicate that rifaximin might upregulate the AQP3 and AQP8 proteins in the colon mucosa, preventing water reabsorption in the lumen by blood vessels and improving constipation. 5-HT and SP are excitatory neurotransmitters and neuropeptides, respectively, associated with the acceleration of intestinal motility [39,40]. In our study, after rifaximin intervention, serum 5-HT and SP were significantly increased in constipated rats, indicating that rifaximin might accelerate colon motility and shorten colonic transit time by upregulating excitatory neurotransmitters or neuropeptides.

Gut microbiota disorders are closely related to the development of constipation, and many studies have shown significant differences in gut microbiota between constipated patients and healthy controls [5,12]. Similarly, the results of 16S rRNA high-throughput sequencing in our study showed significant differences in the gut microbiota between constipated and normal rats, and rifaximin could regulate the gut microbiota in constipated rats. In our study, the Chao index in the RIF group was lower than that in the CTR and LOP groups, indicating that rifaximin might inhibit the growth of some potentially pathogenic bacteria. Although there was no significant difference in the Shannon indices between the three groups, the abundance of gut microbiota had specific alterations at the phylum, genus, and species levels. Rifaximin intervention increased *Akkermansia muciniphila*, which was consistent with the study of Vladimir et al. [41]. Studies have demonstrated that *Akkermansia muciniphila* can promote the host’s intestinal 5-HT biosynthesis, enhance gastrointestinal motility, and improve bowel function [42], which was consistent with the results of our study showing that the concentration of 5-HT and gastrointestinal motility were increased after rifaximin intervention in constipated rats. In addition, the abundance of *Bifidobacterium pseudolongum* was reduced after rifaximin intervention. *Bifidobacterium pseudolongum* is the most frequent species in the animal gut [43]. Interestingly, Tatsuoka et al. [44,45] proposed that *Bifidobacterium pseudolongum* could reduce 5-HT content in the colonic mucosa by diminishing EC cells. *Akkermansia muciniphila* showed significantly positive correlations with AQP3, AQP8, 5-HT, and SP, while *Bifidobacterium pseudolongum* was negatively correlated with AQP3 and AQP8 but positively correlated with TLR2. Furthermore, we found that *Lactobacillus murinus* in constipated rats was significantly decreased compared to that in control rats, which was consistent with the results of Yang et al. [46], and this alteration was reversed by rifaximin. Previous studies have shown that Lactobacillus reuteri and Lactobacillus plantarum were effective in improving functional constipation in children and constipated mice [47,48], indicating that some Lactobacillus species could alleviate constipation.

Bile acids (BAs) are synthesized in the liver, and up to 95% of intestinal bile acids are reabsorbed by ileal bile acid transporters located in the terminal ileum and returned to the liver via the portal vein; only small amounts of bile acids spill over to the colon [49]. BAs might participate in the pathophysiology of constipation; patients with diarrhea had higher levels of BAs in feces than those with constipation [50], and an increase in fecal bile acids is significantly associated with accelerated colonic transit [51,52]. BAs can stimulate defecation by increasing the permeability of the intestinal mucosa and promoting the secretion of water in the intestinal lumen [53]. An inhibitor of the ileal bile acid transporter could alleviate constipation by inhibiting the reabsorption of bile acids and increasing the concentrations of BAs in the colon lumen [54,55]. In this study, we detected serum metabolites using untargeted metabolomic analysis through LC–MS/MS and found that 21 of the 28 BAs detected using untargeted metabolomics were increased in the LOP group compared to the CTR group. The intervention of rifaximin suppressed the increase in the majority of BAs that were increased in the LOP group, which indicates that, compared to control rats, more BAs from the intestines of constipated rats were reabsorbed into the blood, fewer BAs were spilled over to the colon, and rifaximin intervention could reduce the reabsorption of BAs. On the one side, the gastrointestinal transit of the constipated rats was significantly lower than in the CTR and RIF groups, indicating that the BAs could stay longer in the ileum in constipated rats, and more BAs could be reabsorbed into the blood. On the other side, the composition of the microbiota can affect the reabsorption of bile acids. The deconjugation of Bacteroides can prevent the reabsorption of conjugated BAs passing through the apical sodium-dependent bile acid transporter (ASBT) in the terminal ileum [56]. In our study, some bacteria, such as *Faecalibaculum rodentium*, *Bifidobacterium pseudolongum*, and *unclassified_g__Allobaculum,* showed significantly positive correlations with most of the BAs, including glycochenodeoxycholic acid, chenodeoxycholic acid, taurochenodeoxycholic acid, deoxycholic acid, and lithocholic acid, indicating that these bacteria might increase the reabsorption of these BAs from the intestinal lumen to the blood circulation system.

There are several limitations in this study. First, we found that the abundances of some gut bacteria, such as *Akkermansia muciniphila*, *Lactobacillus murinus*, and *Bifidobacterium pseudolongum,* were altered in constipated rats, and we did not further investigate the roles of these altered bacteria in the development of constipation. Second, in constipated rats, more BAs from the intestine were reabsorbed into the blood, but the mechanism that affects reabsorption is not clear. Finally, we measured metabolites in the serum and found an increase in BAs in the serum of constipated rats; however, due to the limitations of fecal sample size, we did not measure the fecal metabolites, and the BAs in feces play a key role in colon motility. More studies are needed in the future to explore the efficacy and mechanism of rifaximin in constipation.

## 5. Conclusions

This study suggested that rifaximin intervention might improve loperamide-induced constipation in rats by increasing intestinal peristalsis, modulating water metabolism, and ameliorating intestinal inflammation. In addition, rifaximin could regulate the gut microbiota of constipated rats, which might affect gastrointestinal motility or the levels of excitatory neurotransmitters, neuropeptides, and other metabolites, such as BAs. BAs and steroid metabolism seem to be an important link in rifaximin treatment of constipation and are worth further study. Based on the multi-omics analysis, the gut microbiota and metabolites might play a vital role in the development and treatment of constipation, among which *Akkermansia muciniphila* and gut-microbiome-related BAs seemed to occupy the most important positions. In the future, more studies are needed to explore the roles of *Akkermansia muciniphila* and gut-microbiome-related BAs in the development and treatment of constipation. In summary, our study found that rifaximin might be a potential strategy to relieve constipation, providing new ideas for the treatment of constipation.

## Figures and Tables

**Figure 1 nutrients-15-04502-f001:**
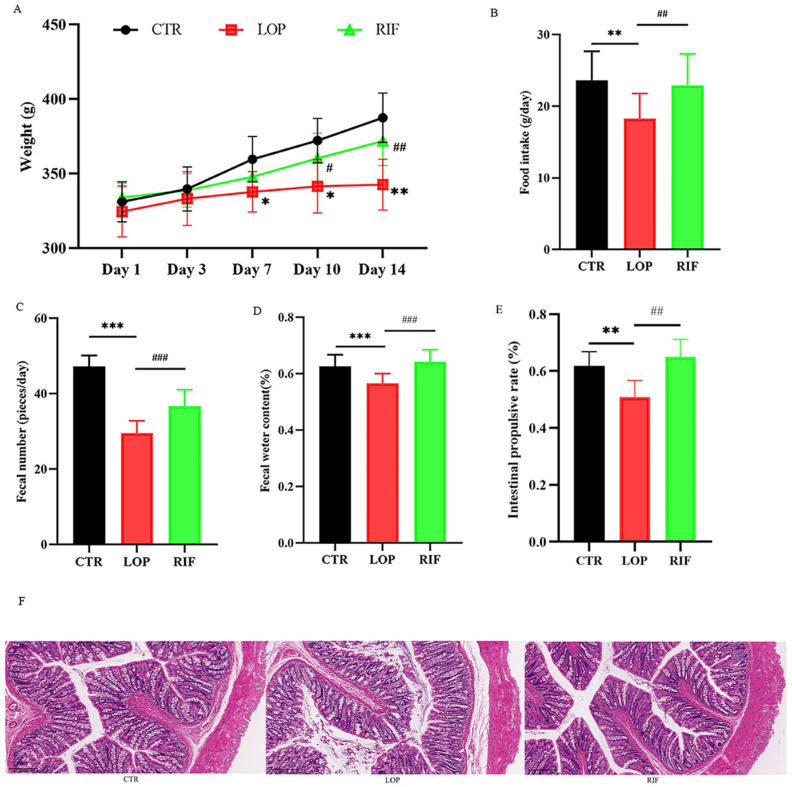
Effects of rifaximin on the constipated indicators and structure of colon tissues in constipated rats. (**A**) Body weight; (**B**) Food intake; (**C**) Fecal number; (**D**) Fecal water content; (**E**) Intestinal propulsive rate; (**F**) H&E staining. Data were represented as mean ± SD. * *p* < 0.05, ** *p* < 0.01, *** *p* < 0.001, LOP vs. CTR group; # *p* < 0.05, ## *p* < 0.01, ### *p* < 0.01, LOP vs. RIF group.

**Figure 2 nutrients-15-04502-f002:**
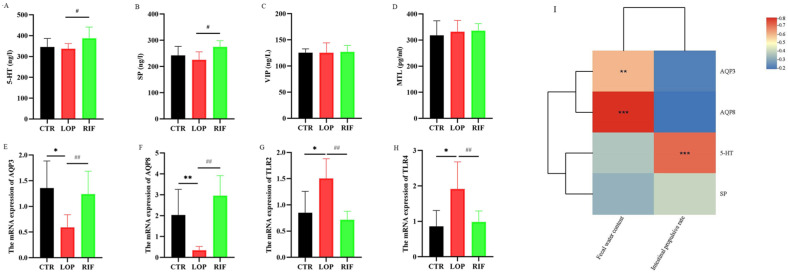
Effects of rifaximin on serum neurotransmitters, neuropeptides, colon inflammation, and water metabolism. (**A**) Serum 5-HT; (**B**–**D**) Serum neuropeptides, including SP, VIP, and MTL; (**E**,**F**) mRNA expression of aquaporins, including AQP3, AQP8; (**G**,**H**) mRNA expression of inflammatory cytokines, including TLR2, TLR4. (**I**) Spearman correlation analysis between aquaporins, neurotransmitter, neuropeptide, and fecal water content, gastrointestinal transit rate in a heatmap. Data were represented as mean ± SD. * *p* < 0.05, ** *p* < 0.01, LOP vs. CTR group; # *p* < 0.05, ## *p* < 0.01, LOP vs. RIF group. In the heatmap of spearman analysis, the color blue indicated a negative correlation and red indicated a positive correlation. Significant correlations were marked by ** *p* < 0.01, *** *p* < 0.001.

**Figure 3 nutrients-15-04502-f003:**
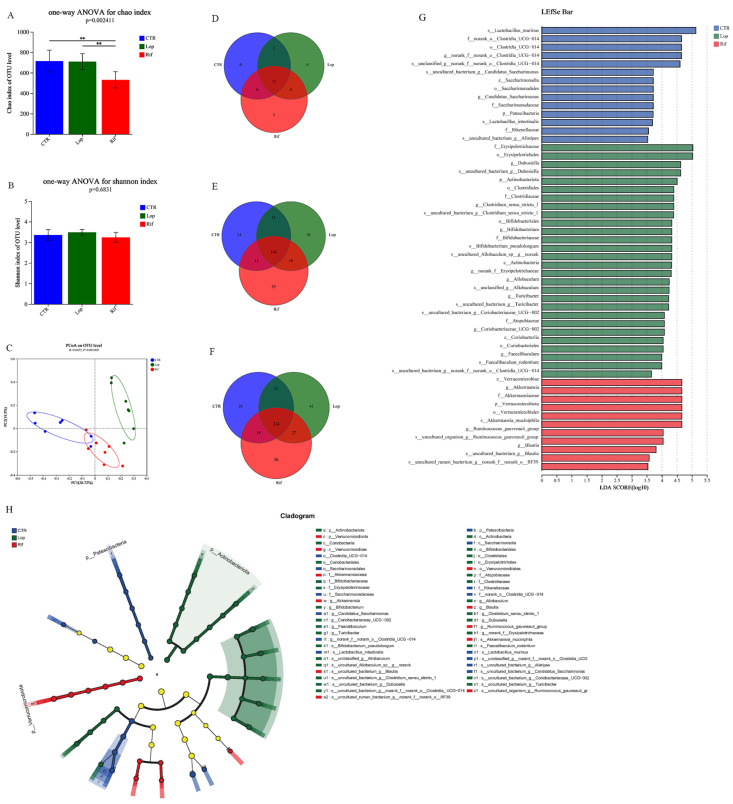
Effects of rifaximin on the structure of gut microbiota. (**A**,**B**) α-diversity index between the three groups. Shannon index (**A**) and Simpson index (**B**). (**C**) β-diversity index between the three groups. Principal coordinate analysis (PCoA) of the microbiota for the three groups. (**D**–**F**) Venn diagram depicting the number of shared and unique bacterial communities among the CTR, LOP, and RIF groups at the levels of phylum (**D**), genus (**E**), and species (**F**). (**G**) Histogram of LDA value distribution generated by LEfSe indicating the differences of bacterial communities among three groups after LDA using a threshold score of LDA > 3.5. (**H**) Cladograms generated using LEfSe indicate the dominant bacteria communities between the three groups after LDA using a threshold score of LDA > 3.5. Data were represented as mean ± SD. ** *p* < 0.01.

**Figure 4 nutrients-15-04502-f004:**
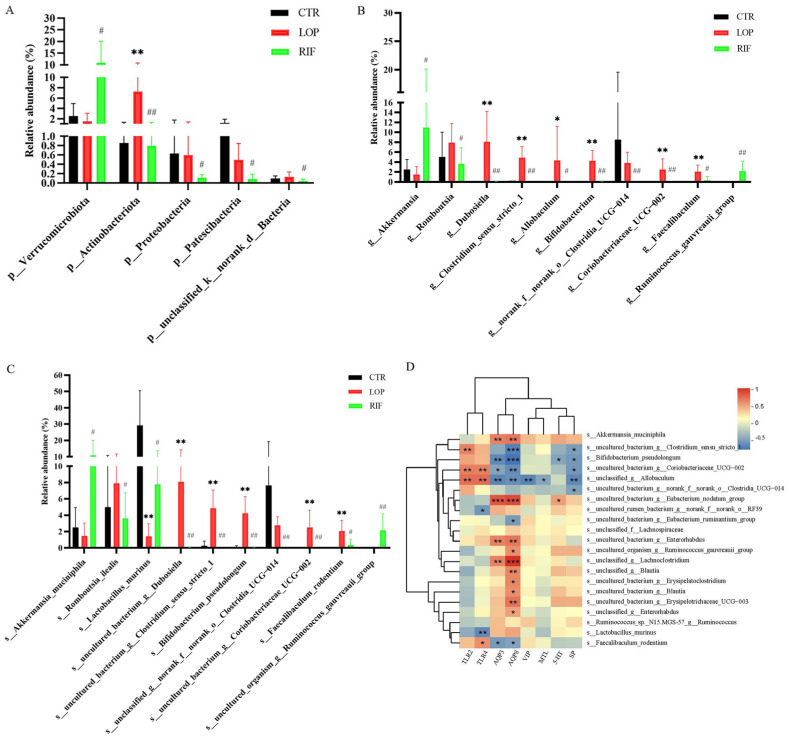
Effects of rifaximin on the structure of gut microbiota in constipated rats. (**A**) Relative abundances of gut microbiota at the phylum level among three groups. (**B**) Relative abundances of gut microbiota at the genus level among three groups. (**C**) Relative abundances of gut microbiota at the special level among three groups. (**D**) Spearman correlation analysis between gut microbiota and neurotransmitters, neuropeptides, aquaporins, and inflammatory cytokines in a heatmap. Data were represented as mean ± SD. * *p* < 0.05, ** *p* < 0.01, LOP vs. CTR group; # *p* < 0.05, ## *p* < 0.01, LOP vs. RIF group. The color blue indicated a negative correlation and red indicated a positive correlation. Significant correlations were marked by * *p* < 0.05, ** *p* < 0.01 and *** *p* < 0.001.

**Figure 5 nutrients-15-04502-f005:**
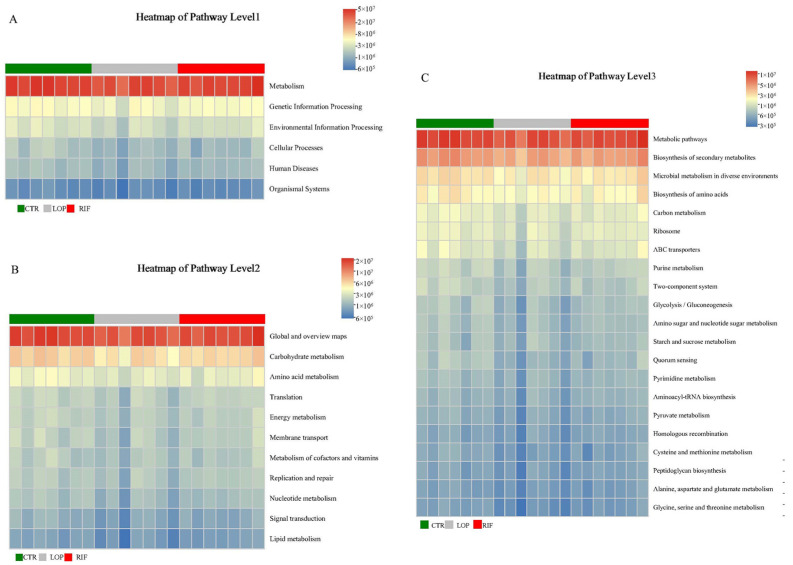
Functional prediction analysis of gut microbiota. (**A**) The functional prediction analysis of gut microbiota based on first-level data information of KEGG. (**B**) The functional prediction analysis of gut microbiota based on second-level data information of KEGG. (**C**) The functional prediction analysis of gut microbiota based on third-level data information of KEGG.

**Figure 6 nutrients-15-04502-f006:**
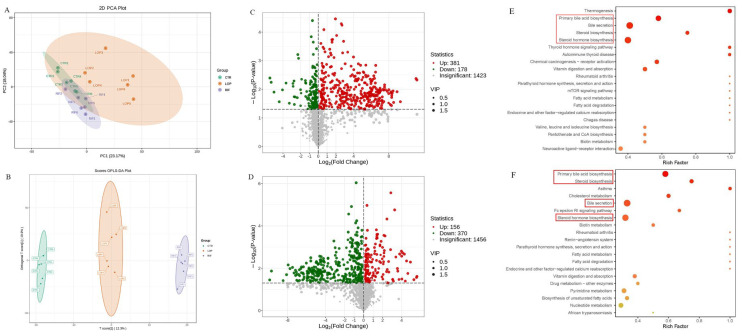
Effects of rifaximin on the serum metabolites in constipated rats. (**A**) Principal coordinate analysis (PCoA) based on all identified metabolites. (**B**) Orthogonal partial least-squares discriminant analysis (OPLS-DA) based on all identified metabolites. (**C**) Volcano plot of differential metabolites between CTR and LOP groups. (**D**) Volcano plot of differential metabolites between LOP and RIF groups. (**E**) KEGG enrichment analysis based on the CTR and LOP groups. (**F**) KEGG enrichment analysis based on the LOP and RIF groups. The red blanks in (**E**,**F**) showed the metabolic pathways that were significantly different among the three groups.

**Figure 7 nutrients-15-04502-f007:**
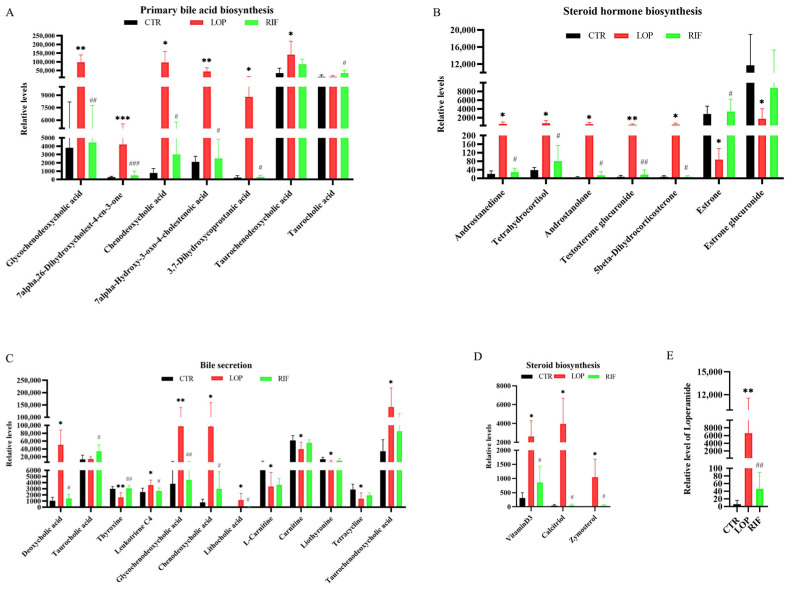
The relative levels of serum metabolites in the KEGG enrichment analysis had significant differences among the three groups. (**A**) The differential metabolites between three groups in primary bile acid biosynthesis. (**B**) The differential metabolites between three groups in steroid biosynthesis. (**C**) The differential metabolites between three groups in bile secretion. (**D**) The differential metabolites between three groups in steroid hormone biosynthesis. (**E**) Relative level of loperamide between three groups. Data were represented as mean ± SD. * *p* < 0.05, ** *p* < 0.01, *** *p* < 0.01, LOP vs. CTR group; # *p* < 0.05, ## *p* < 0.01, ### *p* < 0.001, LOP vs. RIF group.

**Figure 8 nutrients-15-04502-f008:**
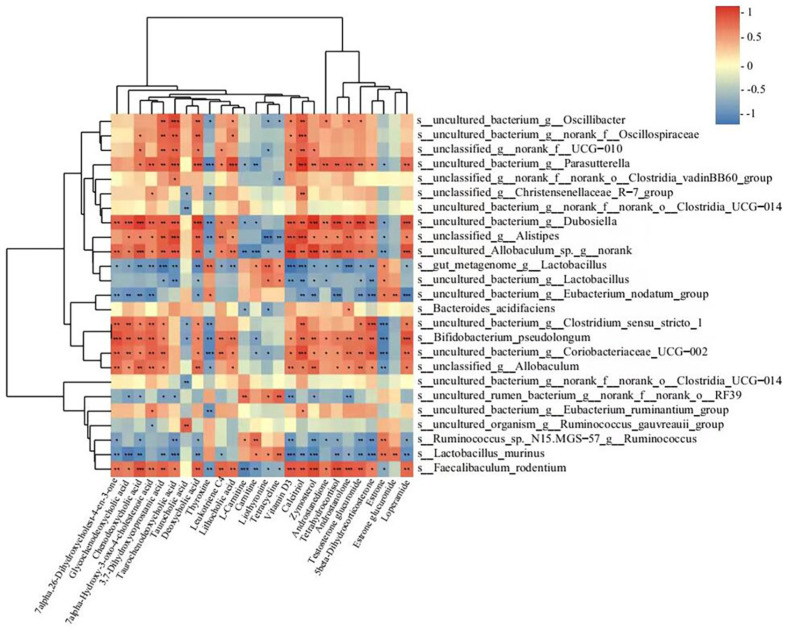
Spearman correlation analysis between fecal microbiota and serum metabolites in a heatmap. The color blue indicated a negative correlation and red indicated a positive correlation. Significant correlations were marked by * *p* < 0.05, ** *p* < 0.01, and *** *p* < 0.001.

**Figure 9 nutrients-15-04502-f009:**
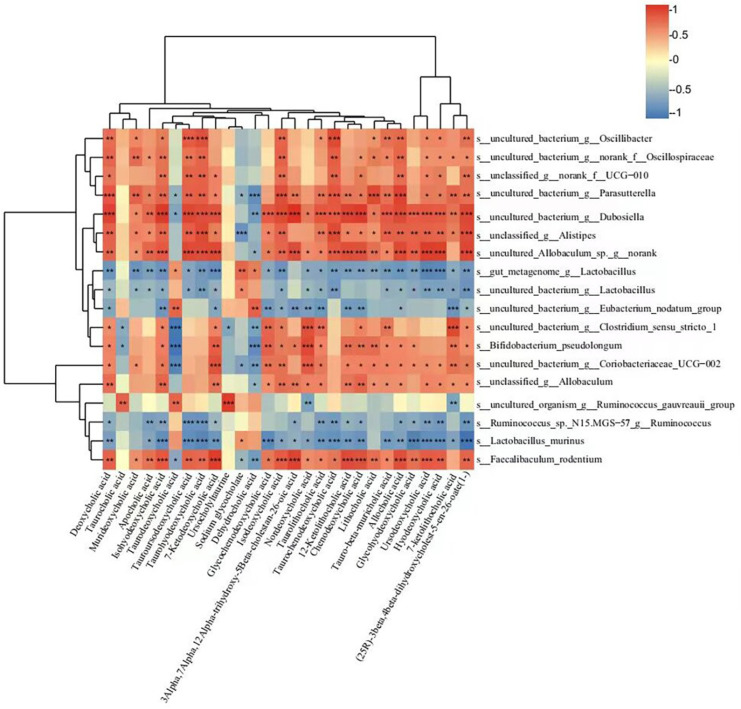
Spearman correlation analysis between gut microbiota and serum BAs in a heatmap. The color blue indicated a negative correlation and red indicated a positive correlation. Significant correlations were marked by * *p* < 0.05, ** *p* < 0.01, and *** *p* < 0.001.

**Table 1 nutrients-15-04502-t001:** The BAs with differences in abundance between CTR and LOP groups.

Compounds name	Mean CTR	Mean LOP	VIP	*p*-Value	Fold Change	Log2FC	Change in LOP
Deoxycholic acid	1066.260	50,606.155	1.633	0.022	47.461	5.569	up
Apocholic acid	11,102.901	106,959.609	1.330	0.031	9.633	3.268	up
Isohyodeoxycholic acid	178.559	16,957.735	1.651	0.012	94.970	6.569	up
Taurodeoxycholic acid	6364.034	970.079	1.634	0.034	0.152	−2.714	down
Tauroursodeoxycholic acid	677.763	19,139.782	1.604	0.045	28.240	4.820	up
Taurohyodeoxycholic acid	3784.726	25,107.742	1.550	0.014	6.634	2.730	up
7-Ketodeoxycholic acid	259.556	165,052.412	1.748	0.027	635.902	9.313	up
Dehydrocholic acid	21,229.988	12,196.397	1.256	0.015	0.574	−0.800	down
Glycochenodeoxycholic acid	3807.062	97,532.070	1.784	0.003	25.619	4.679	up
Isodeoxycholic acid	3700.985	202,609.462	1.712	0.017	54.745	5.775	up
3Alpha,7Alpha,12Alpha-trihydroxy-5Beta-cholestan-26-oic acid	2776.560	20,266.751	1.777	0.000	7.299	2.868	up
Nordeoxycholic acid	456.004	7336.268	1.574	0.020	16.088	4.008	up
Taurochenodeoxycholic acid	34,228.022	141,577.639	1.440	0.017	4.136	2.048	up
12-Ketolithocholic acid	1168.491	43,504.507	1.654	0.024	37.231	5.218	up
Chenodeoxycholic acid	783.305	96,992.442	1.766	0.013	123.825	6.952	up
Lithocholic acid	30.397	1182.747	1.304	0.047	38.910	5.282	up
Tauro-beta-muricholic acid	22.559	1594.160	1.710	0.009	70.667	6.143	up
Allocholic acid	18.281	1759.077	1.555	0.016	96.226	6.588	up
Glycohyodeoxycholic acid	7268.141	44,534.381	1.651	0.000	6.127	2.615	up
Ursodeoxycholic acid	35,044.860	466,779.821	1.573	0.035	13.319	3.735	up
Hyodeoxycholic acid	25,480.018	360,940.644	1.522	0.034	14.166	3.824	up
7-ketolithocholic acid	61.692	4127.493	1.655	0.016	66.905	6.064	up
(25R)-3beta,4beta-dihydroxycholest-5-en-26-oate(1-)	10.766	1369.281	1.723	0.012	127.187	6.991	up

VIP: Variable Importance in Projection; FC: Fold Change.

**Table 2 nutrients-15-04502-t002:** The BAs with differences in abundance between LOP and RIF groups.

Compounds Name	Mean LOP	Mean RIF	VIP	*p*-Value	Fold Change	Log2FC	Change in RIF
Deoxycholic acid	50,606.155	1425.796	1.650	0.022	0.028	−5.149	down
Taurocholic acid	14,283.205	34,218.129	1.465	0.025	2.396	1.260	up
Isohyodeoxycholic acid	16,957.735	699.985	1.592	0.014	0.041	−4.598	down
Taurodeoxycholic acid	970.079	18,803.560	1.862	0.002	19.384	4.277	up
7-Ketodeoxycholic acid	165,052.412	3830.114	1.597	0.029	0.023	−5.429	down
Ursocholyltaurine	20,896.284	66,936.404	1.734	0.005	3.203	1.680	up
Sodium glycocholate	5364.930	6095.290	1.097	0.050	1.136	0.184	up
Dehydrocholic acid	12,196.397	26,944.344	1.494	0.002	2.209	1.144	up
Glycochenodeoxycholic acid	97,532.070	4435.374	1.839	0.003	0.045	−4.459	down
Isodeoxycholic acid	202,609.462	4877.642	1.736	0.018	0.024	−5.376	down
Nordeoxycholic acid	7336.268	79.043	1.891	0.016	0.011	−6.536	down
12-Ketolithocholic acid	43,504.507	3121.753	1.528	0.029	0.072	−3.801	down
Chenodeoxycholic acid	96,992.442	2994.280	1.676	0.014	0.031	−5.018	down
Lithocholic acid	1182.747	24.644	1.442	0.046	0.021	−5.585	down
Tauro-beta-muricholic acid	1594.160	29.811	1.771	0.010	0.019	−5.741	down
Allocholic acid	1759.077	72.740	1.385	0.018	0.041	−4.596	down
Glycohyodeoxycholic acid	44,534.381	21,644.455	1.270	0.012	0.486	−1.041	down
7-ketolithocholic acid	4127.493	30.843	1.767	0.015	0.007	−7.064	down
(25R)-3beta,4beta-dihydroxycholest-5-en-26-oate(1-)	1369.281	33.189	1.669	0.013	0.024	−5.367	down

VIP: Variable Importance in Projection; FC: Fold Change.

## Data Availability

The datasets used and/or analyzed during the current study are available from the corresponding author upon reasonable request.

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
