# Peer review of "Rifaximin Ameliorates Loperamide-Induced Constipation in Rats through the Regulation of Gut Microbiota and Serum Metabolites"

_nutrients, 2023, doi:10.3390/nu15214502_

Round 1

Reviewer 1 Report

 Dear Esteemed Authors of Manuscript Nutrients-2659610,

I am writing to express my sincere gratitude for the privilege of being entrusted with the role of peer reviewer for your esteemed paper. It is indeed an honor that Nutrients has bestowed upon me, and I assure you of my utmost dedication to impartially evaluate the evidence presented in your manuscript.

I kindly request the esteemed authors to carefully consider and address the constructive commentaries provided below. I believe that incorporating these suggestions will undoubtedly elevate the overall quality and impact of your valuable contribution.

Wishing you continued success in your research endeavors.

Warm regards,

The Peer Reviewer

---

INTRODUCTION

1) I would like to kindly propose the inclusion of a dedicated paragraph providing crucial insights into the antibiotic Rifaximin. This addition should encompass essential details such as its classification within the antibiotic spectrum and its prevalent applications in clinical practice. This enhancement would undoubtedly serve to enrich the depth and comprehensiveness of your manuscript.

2) I suggest that the author incorporates a dedicated paragraph within the introduction section, delving into the intricacies of gut dysbiosis pathogenesis. This would serve to provide a comprehensive foundation for readers, offering them a nuanced understanding of the underlying processes driving this condition.

3) In addition to the aforementioned suggestion, I recommend the incorporation of an additional dedicated paragraph focused on prior studies that have employed Rifaximin in the treatment of gut dysbiosis. Within this section, it would be beneficial to succinctly outline each of the selected studies, elucidating their methodologies and key findings. Furthermore, provide a comprehensive discussion on bacterial transformation resultant from Rifaximin administration, while also delving into the clinical implications derived from these investigations. Emphasize how these discoveries may hold significant promise for addressing functional constipation, thereby augmenting the potential impact of your research. In this context, elucidate the specific reasons why your study holds greater relevance compared to previously published works.

MATERIALS AND METHODS

4) Please furnish detailed explanations regarding the selection of specific statistical tests employed in your study, along with the underlying rationale for their choice. Additionally, elaborate on which dataset was utilized for calculating means and which dataset was used for determining standard deviations. This elucidation will fortify the methodological framework of your research, ensuring a thorough comprehension for readers.

5) Within sections 2.5. and 2.6.1, when conducting the analysis of RNA sequencing, it is imperative to furnish explicit details regarding the concentrations of the reagents utilized. This information is crucial for the replication and validation of your experimental procedures, ensuring transparency and reproducibility in your research methodology.

6) At lines 84-85, please furnish the complete form of the equation. Subsequently, immediately present the current mathematical expression in its entirety for clarity and reference. This dual approach will afford readers a comprehensive understanding of the equation's structure and facilitate accurate interpretation.

7) Within section 2.2.1, it is requested to provide a more detailed account of the procedures concerning the weighting of the rats and any observed alterations in their food intake. Additionally, please include specific information about the type or model of the scale used, particularly mentioning the scale mark. This additional information will enhance the clarity and replicability of your experimental methodology. Specify the time of day when the measurements were taken, along with the environmental conditions. Were the rats prevented from eating or drinking for a certain period before the measurement?

DISCUSSION

8) While acknowledging the limitations outlined in your study's discussion, it is imperative to underscore the significant strengths it possesses. This can be achieved through the introduction of a dedicated paragraph in the discussion section, illuminating the distinctive merits of your research. Moreover, it is pivotal to conduct a comparative analysis, juxtaposing the strengths of your study against the acknowledged limitations of previously published works, thereby establishing its unique contributions to the academic discourse.

CONCLUSION

9) Kindly integrate a comprehensive concluding section into this manuscript, demonstrating a nuanced exploration of potential future research directions. Emphasize the paramount discoveries unearthed throughout this study and illuminate their far-reaching implications for the future of the field. Delve into the promising avenues that emerge from these findings, offering a forward-looking perspective that can inspire and guide scholars in their pursuit of new knowledge and innovation. Place special emphasis on showcasing the potential of advanced delivery systems in seamlessly incorporating antibiotics into the gastrointestinal tract.

The manuscript exhibits a commendable overall merit, showcasing substantial promise. Nonetheless, there exist certain limitations that warrant careful consideration and resolution prior to its publication. These constraints, once effectively addressed, will undoubtedly bolster the manuscript's scholarly impact and comprehensiveness.

Author Response

Please see the attachment. Due to the limitation of file data size (<120MB), we placed the high-resolution images in the Figures file.

Reviewer 2 Report

xiong et al. submitted the manuscript entitled Rifaximin ameliorates loperamide-induced constipation in rats through the regulation of gut microbiota and serum metabolites, in which they found Rifaximin may be able to potentially used for the treatment of constipation. They analyzed different factors and expression change of neurotransmitters, neuropeptides, mRNA as well as metabolite change in gut sample suffering from constipation. The authors also tried to deduce how these factors function in the progress of  constipation based on the data. From the text of manuscript I can primarily tell this is a good work. But the figures in the manuscript are really in low quality. Since the conclusion highly relies on the figures, I can only evaluate this work after the authors provide with high-resolution figures.

Besides, I have one concern for section 2.1: the authors tried to build a constipation with administration of loperamide. Please provide 1. reference for this model and 2. what are the indicators to prove the model is successfully established?

Author Response

(The authors gave the same response as above.)

Reviewer 3 Report

The research by Luo, et.al investigated how rifaximin, an antibiotic known for positively regulating the gut microbiota, exerts the pharmacology effect against constipation. Using rat model with loperamide-induced constipation, the study found that rifaximin alleviated constipation symptoms. The treatment of rifaximin increased levels of serum 5-HT, SP, and specific mRNA expressions, while decreasing others. Notably, the antibiotic regulated the gut flora by boosting beneficial bacteria levels. Furthermore, the metabolomics analysis showed shifts in serum metabolites in constipated rats, which were reversed with rifaximin treatment. Conclusively, rifaximin might be a promising therapeutic approach for constipation, as it can modulate neurotransmitters, neuropeptides, water metabolism, and reduce intestinal inflammation, all by regulating the gut microbiota and serum metabolites. The manuscript is overall in good logic and the conclusion can be supported by the assay results. I have some minor comments for authors’ consideration

(1)   Please make sure all figures have the resolution > 300 dpi.

(2)   Could author specify whether the bowel movement (GI transit) would significantly be slower in the constipation condition? And how rifaximin can accelerate the bowel movement. Is the increased neurotransmitter as the pharmacology mechanism underlying?

(3)   In line 494, the authors stated that 21 of 28 BAs detected by untargeted metabolomics were increased in the LOP group. Given the bile acid is primarily re-absorbed in ileum (ileum exhibit a much larger surface area for absorption than colon), the slowing GI transit (enabling BA staying longer in ileum) may be another mechanism to explain the increased BA reabsorption in LOP group. 

none

Author Response

(The authors gave the same response as above.)

Reviewer 4 Report

Rifaximin ameliorates loperamide-induced constipation in rats through the regulation of gut microbiota and serum metabolites presents an interesting study about the effect of rifaximin on loperamide-induced constipation in rats. However, before being accepted for publication, I believe that some improvements to the work are necessary. My suggestions are listed below:

Introduction typically includes additional elements to set the stage for the research. Here are some suggestions for enhancing the introduction:

- Highlight the existing knowledge gap or unanswered questions in the field of constipation research, particularly in relation to its pathogenesis and treatment.

- Emphasize the need for novel strategies to address constipation, considering the complexity of its pathogenesis and the potential role of gut microbiota.

- Explain the rationale for considering rifaximin as a potential treatment option for constipation, based on its effects on gut microbiota.

Figure 1 does not have a very good resolution - it would be ideal if you could replace it with another image at a higher resolution - Also valid for the other figures, although I zoomed in, I could not read anything from the writing on the figures, and it is a shame that the work done is not properly presented.

The text mentions that concentrations of loperamide, which was used to induce constipation, were downregulated in the RIF group compared to the LOP group. Could you explain the significance of this finding and how it relates to the overall effects of rifaximin on constipation and metabolite pathways?

Given the observed alterations in the gut microbiota composition, particularly the increase in Akkermansia muciniphila and the decrease in Bifidobacterium pseudolongum and Lactobacillus murinus after rifaximin intervention, could you further elucidate the specific mechanisms through which these changes in gut microbiota contribute to the improvement of constipation in rats? Additionally, are there any hypotheses or theories regarding the interplay between these microbial species and host factors, such as 5-HT, AQP3, AQP8, and TLR2, that may explain their impact on bowel function and gastrointestinal motility?

Author Response

(The authors gave the same response as above.)

Round 2

Reviewer 1 Report

The quality of the manuscript has undergone a substantial improvement, a testament to the diligent efforts of the esteemed authors. I extend my heartfelt gratitude for your dedication. Based on the remarkable progress, I wholeheartedly endorse the publication of your manuscript, confident that it will make a valuable contribution to the academic community.

Some minor adjustments in English are necessary. I have confidence that MDPI's proficient team will diligently attend to all concerns during the final English-editing phase following the peer review process. This ensures that the manuscript meets the highest standards of clarity and coherence.

Reviewer 2 Report

The overall quality of this manuscript is good. But the resolution of figure 3 and figure 6 need further improvement. If the authors are limited to file data size, they can consider putting the corresponding figures in supplemental data.

I have no other concerns if authors can address on the figure issue.